# Effect of Sm Doping on the Microstructure, Mechanical Properties and Shape Memory Effect of Cu-13.0Al-4.0Ni Alloy

**DOI:** 10.3390/ma14144007

**Published:** 2021-07-17

**Authors:** Qimeng Zhang, Bo Cui, Bin Sun, Xin Zhang, Zhizhong Dong, Qingsuo Liu, Tianyu Cui

**Affiliations:** 1School of Materials Science and Engineering, Tianjin University of Technology, Tianjin 300384, China; qm203170331@126.com (Q.Z.); zhizhong.dong@email.tjut.edu.cn (Z.D.); 2Institute of Nuclear Physics and Chemistry, China Academy of Engineering Physics, Mianyang 621900, China; cuibo13@163.com; 3School of Material Science and Chemical Engineering, Harbin Engineering University, Harbin 150001, China; sunbin2040@163.com; 4Beijing Advanced Innovation Center for Materials Genome Engineering, Institute for Advanced Materials and Technology, University of Science and Technology Beijing, Beijing 100083, China; b20190604@xs.ustb.edu.cn

**Keywords:** Cu-Al-Ni, high temperature shape memory alloy, mechanical property, shape memory effect

## Abstract

The effects of rare earth element Sm on the microstructure, mechanical properties, and shape memory effect of the high temperature shape memory alloy, Cu-13.0Al-4.0Ni-*x*Sm (*x* = 0, 0.2 and 0.5) (wt.%), are studied in this work. The results show that the Sm addition reduces the grain size of the Cu-13.0Al-4.0Ni alloy from millimeters to hundreds of microns. The microstructure of the Cu-13.0Al-4.0Ni-*x*Sm alloys are composed of 18R and a face-centered cubic Sm-rich phase at room temperature. In addition, because the addition of the Sm element enhances the fine-grain strengthening effect, the mechanical properties and the shape memory effect of the Cu-13.0Al-4.0Ni alloy were greatly improved. When *x* = 0.5, the compressive fracture stress and the compressive fracture strain increased from 580 MPa, 10.5% to 1021 MPa, 14.8%, respectively. When the pre-strain is 10%, a reversible strain of 6.3% can be obtained for the Cu-13.0Al-4.0Ni-0.2Sm alloy.

## 1. Introduction

Shape memory alloys (SMAs) are functional metal materials. Due to its reversible thermoelastic martensitic transformation [1,2,3], SMAs have very special mechanical properties, such as hyperelasticity and shape memory effect (SME), making them suitable for sensing and drive applications [4]. The martensitic transformation temperature of some developed SMAs such as Ni-Ti is less than 100 °C [5,6,7], limiting their further application. Therefore, more and more studies are focusing on the application of SMAs at higher temperatures, and in particular, high temperature shape memory alloys (HTSMAs) have the potential to be used as solid-state actuators in high temperature fields of aerospace, nuclear power, fire, oil, and gas exploration [8,9,10,11,12,13,14].

At present, the Cu-Al-Ni alloy has become a potential HTSMA due to the low cost and outstanding properties of its single crystal [15,16,17]. However, the severe brittleness of polycrystalline Cu-Al-Ni alloys limits its practical application, which is related to its large elastic anisotropy and large grain size [10,18,19]. In recent years, powder metallurgy, rapid solidification, and alloying methods are all methods used for improving the mechanical properties and mechanical properties of alloys by reducing the grain size [17,20,21]. Among them, the alloying method has the characteristic of simple equipment, and so the addition of the fourth element is considered to be an effective method to improve the mechanical properties of Cu-Al-Ni SMAs, with simple operation and convenient production [22,23,24,25]. Recently, some predecessors have conducted studies on adding Ti, B, Be, Mn, Ge, Co, and V elements to Cu-Al-Ni alloys [16,26,27,28,29,30,31,32]. For example, the addition of Co can effectively increase the tensile fracture strength of the Cu-11.9Al-4.0Ni alloy from 270 to 650 MPa, and the tensile fracture strain can increase from 1.65 to 7%. In addition, studies have proven that the addition of rare earth elements can change the microstructure and mechanical properties of Cu-Al-Ni SMAs. We previously conducted research on adding rare earth elements such as Gd and Nd to the Cu-13.0Al-4.0Ni alloy [33,34]. The results show that the addition of rare earth elements greatly improves the mechanical properties of the Cu-13.0Al-4.0Ni alloy. At present, there are no related reports and studies on the doping of rare earth element Sm in Cu-Al-Ni alloys. Therefore, the current paper aims to investigate the effects of various additions of Sm on the structure, mechanical properties, and SME of the Cu-13.0Al-4.0Ni alloy.

## 2. Materials and Methods

In this experiment, 99.99% pure metal particles of Cu, Al, Ni, and Sm purchased from Beijing Hawk Science and Technology Co., Ltd. (Beijing, China) were selected for preparing the Cu-13.0Al-4.0Ni-*x*Sm (*x* = 0.2, 0.5) (wt.%) alloy used in the experiment. The alloy was first melted with a non-consumed vacuum-arc melting furnace under the protection of argon. The raw materials were melted 8 times, then homogenized at 850 °C for 24 h, and finally quenched in ice water for the purpose of melting more uniformly.

First, the alloys were cut into an 8 mm × 8 mm × 2 mm square sample, and then X-ray diffraction was performed (Rigaku D/max-rB XRD, Tokyo, Japan, with Cu Kα radiation) after polishing, at a sweep rate of 4°/min. Second, polish the sample and observe the optical microstructure of the sample with a scanning electron microscope (ZEISS MERLIN Compact SEM, Ober-kochen, Germany). 

The sample used for TEM observation was electropolished in a solution consisting of 2.5 g FeCl_3_·6H_2_O (CAS: 10025-77-1, Tianjin Fuchen Chemical Reagent Factory, Tianjin, China), 10 mL HCl (36.0–38.0 wt.%, CAS: 7647-01-0, Tianjin Fengchuan Chemical Reagent Technology Co., Ltd., Tianjin, China), and 48 mL CH_3_OH (≥99.5%, CAS: 64-56-1, Tianjin Benchmark Chemical Reagent Co., Ltd., Tianjin, China). FEI TECNAI G^2^ 20 STWIN 200 kV TEM equipped with a double-tilt cooling stage and an energy dispersive spectrometer (EDS) was used for transmission observation and surface composition analysis of samples. The phase transition temperature of the alloy was measured by differential thermal analysis (DTA) using the EXSTAR6000, and the heating/cooling rate during the test was 10 °C/min.

The SME and mechanical performance test used cylindrical samples with a size of Ø 3 mm × 5 mm. The samples were subjected to 8% and 10% pre-strain and compression fracture experiments, respectively, performed on the GTN50 electronic universal testing machine. The specific measurement method and calculation formula are the same as those in our previous article [33].

## 3. Results and Discussion

Figure 1a,b shows the metallographic photo of solution treated Cu-13.0Al-4.0Ni-*x*Sm alloys at room temperature. It can be seen that the grain size is decreased obviously with the Sm content increased. The grain size of the Cu-13.0Al-4.0Ni alloy is between 1 and 3 mm, and the size reached the millimeter level [34]. When the Sm content is 0.2 wt.%, the average grain size is about 300 μm. When the Sm content is increased to 0.5 wt.%, the grain size of the alloy is between 100 and 300 μm. The grain size is significantly refined compared with the alloy without Sm. Figure 1c,d shows the SEM micrograph of Cu-13.0Al-4.0Ni-*x*Sm (*x* = 0.2, 0.5) alloys. It can be observed that there are fine bamboo-shaped 18R martensite in the matrix. With the addition of the Sm element, the particles of the second phase are formed.

Figure 2 shows the XRD results of Cu-13.0Al-4.0Ni-*x*Sm (*x* = 0, 0.2, and 0.5) alloys. The peaks of (1 2 2), (2 0 2), (0 0 18), (1 2 8), (1 2 10), (2 0 10), (0 4 0), and (3 2 0) belong to the monoclinic 18R martensite. In addition, several additional peaks of (2 2 2), (4 2 2), (5 2 1), (6 1 1), and (6 3 3) were observed, which correspond to the face-centered cubic phase. The intensity of the diffraction peak increases with the increase of the Sm content, indicating that the number of the second phase increases.

In order to further analyze the phase composition of the Cu-13.0Al-4.0Ni-*x*Sm alloy, TEM observation was made on the Cu-13.0Al-4.0Ni-0.5Sm alloy. Figure 3a is a bright field image of the Cu-13.0Al-4.0Ni-*x*Sm alloy. It can be seen that there is a second phase embedded in the matrix with a grain size of about 400 nm in the alloy. Figure 3b,c shows the electron diffraction patterns of the selected areas in Figure 3a; it can be indexed that the matrix is 18R martensite with a monoclinic structure, and the second phase has a clear face-centered cubic structure. In addition, EDS was performed on the second phase, and the analysis results of Cu, Al, Ni, and Sm are 74.00, 13.68, 2.92, and 9.38 at.%, respectively. Combined with the EDS surface scan images of Cu, Al, Ni, and Sm (Figure 3d–g), it can be seen that the second item in the sample is a Sm-rich phase with a face-centered cubic structure.

Figure 4 is the DTA curve of Cu-13.0Al-4.0Ni-*x*Sm (*x* = 0.2, 0.5) alloys. It can be seen that the addition of rare earth Sm element can significantly reduce the martensitic transformation temperature of the Cu-13.0Al-4.0Ni alloy. The austenite transformation start temperature (*A_s_*), the austenite transformation finish temperature (*A*_f_), the martensitic transformation start temperature (*M_s_*), and the martensitic transformation finish temperature (*M_f_*) are listed in Table 1. The reason for the decrease of the martensitic transformation temperatures may be due to the addition of the Sm element; the formation of the Sm-rich phase increases the content of Al in the matrix, resulting in a decrease of the martensitic transformation temperature. Adding Ti, B, Be, Mn, Ge, Gd, Nd, and Sm elements reduces the martensite transformation temperature of Cu-Al-Ni SMAs [16,27,28,29,33,34]. Among them, the Cu-Al-Ni SMAs doped with Sm has the greatest degree of decrease in the martensite transformation temperature.

Figure 5 is the compressive stress–strain curve of the Cu-13.0Al-4.0Ni-*x*Sm (*x* = 0.2, 0.5) alloys at room temperature. The compressive fracture strength and compressive fracture strain of the Cu-13.0Al-4.0Ni alloy are 580 MPa and 10.5%, respectively [33]. When the Sm content is 0.2%, the compressive fracture strength and the compressive fracture strain of the sample increase to 830 MPa and 13.9%, respectively. With the Sm content gradually increasing to 0.5%, the compressive fracture strength and the compressive fracture strain of the sample also further increased to 1021 MPa and 14.8%, respectively. The mechanical properties improvement is mainly ascribed to the grain refinement, which limits the movement of internal dislocations in the alloy. In order to compare the effects of different doping elements on the mechanical properties of polycrystalline Cu-Al-Ni alloys, the mechanical properties of alloys with similar compositions and test methods are listed in Table 2. It can be seen that although the strength improvement of the Cu-Al-Ni alloys with the addition of Sm element is smaller than that with the addition of B, Ce, and V elements, it is still better than the addition of Mn, Ge, Te, Gd and Nd elements. In the improvement of the alloy’s plasticity, the effect of the doped Sm element is similar to that of the doped B element; while it is not as good as the doped Mn, V, Gd, and Nd elements, it is still better than the doped Ge, Ce and Te elements [26,28,30,32,33,34].

Figure 6 is the stress–strain recovery curve of the Cu-13.0Al-4.0Ni-Sm (*x* = 0.2, 0.5) alloys under pre-strain of 8 and 10%. The arrows indicate the SME after heating at 350 °C for 1 min. It can be seen that under an 8% pre-strain and heating to 350 °C for 1 min, the reversible strain of Cu-13.0Al-4.0Ni-0.2Sm and Cu-13.0Al-4.0Ni-0.5Sm are 5.4 and 3.7%, respectively. These values are higher than the 2.6% [33] of the Cu-13.0Al-4.0Ni alloy, mainly because of the improved mechanical properties of the alloys. When the pre-strain is 10%, the reversible strain of Cu-13.0Al-4.0Ni-0.2Sm and Cu-13.0Al-4.0Ni-0.5Sm are 6.3% and 5.0%, respectively. When *x* = 0.5, more Sm-rich phases are formed, leading to the SME being suppressed, and the less reversible strain is obtained. Compared with most non-rare earth elements, the rare earth element Sm can better improve the SME of the Cu-Al-Ni alloys, but its effect is not as good as that of the B element and the rare earth elements Gd and Nd [26,33,34].

## 4. Conclusions

With the increase of Sm content, the grains of the Cu-13.0Al-4.0Ni alloy were significantly refined, and the internal phase composition of the Cu-13.0Al-4.0Ni alloy also changed. The addition of Sm causes the 2H martensite in the Cu-13.0Al-4.0Ni alloy to disappear and become a single-phase 18R martensite, which is accompanied by the formation of a Sm-rich second phase in the process. In addition, due to fine-grain strengthening, the mechanical properties and SME of the Cu-13.0Al-4.0Ni alloy are greatly improved.

However, the rare earth element Sm cannot be added to the Cu-Al-Ni alloys without limitation, because as the Sm content in the alloy increases, the number of the Sm-rich second phase will also increase. In addition, the influence of a Sm-rich second phase on the mechanical properties of the Cu-Al-Ni alloys is not clear, and thus an in-depth study is needed in the future, but a Sm-rich second phase can definitely lower the SME of the alloy.

## Figures and Tables

**Figure 1 materials-14-04007-f001:**
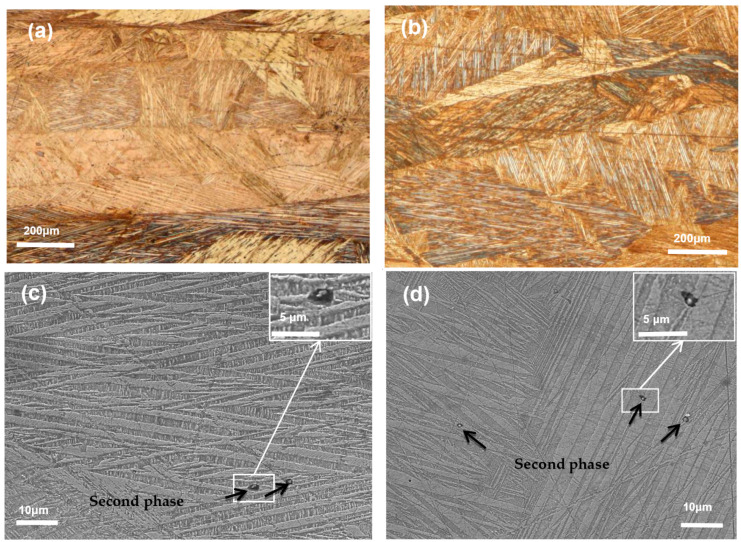
Metallographic photo of Cu-13.0Al-4.0Ni-*x*Sm (*x* = 0.2 (**a**), 0.5 (**b**)) alloys and SEM micrograph of Cu-13.0Al-4.0Ni-*x*Sm (*x* = 0.2 (**c**), 0.5 (**d**)) alloys, the illustration in (**c**,**d**) is a partial enlarged view of the box part.

**Figure 2 materials-14-04007-f002:**
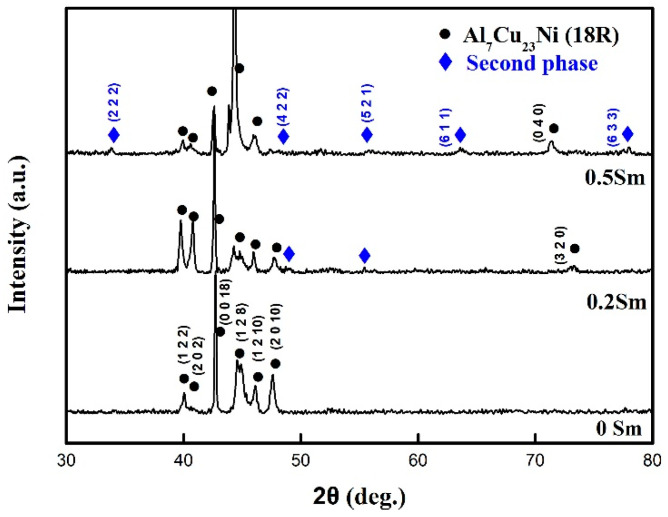
XRD of Cu-13.0Al-4.0Ni-*x*Sm (*x* = 0, 0.2 and 0.5) alloys.

**Figure 3 materials-14-04007-f003:**
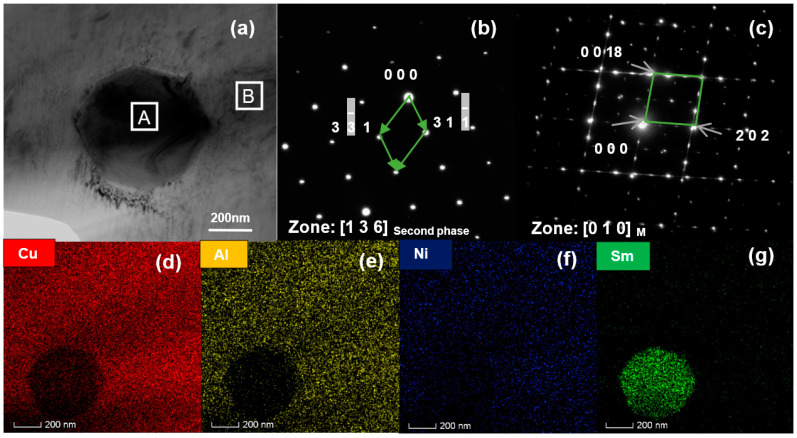
(**a**) The transmission photograph of the second phase in the Cu-13.0Al-4.0Ni-0.5Sm alloy; (**b**) the electron diffraction of the selected area A in (**a**); (**c**) the electron diffraction of the selected area B in (**a**); (**d**–**g**) EDS surface scan pictures of Cu, Al, Ni, and Sm.

**Figure 4 materials-14-04007-f004:**
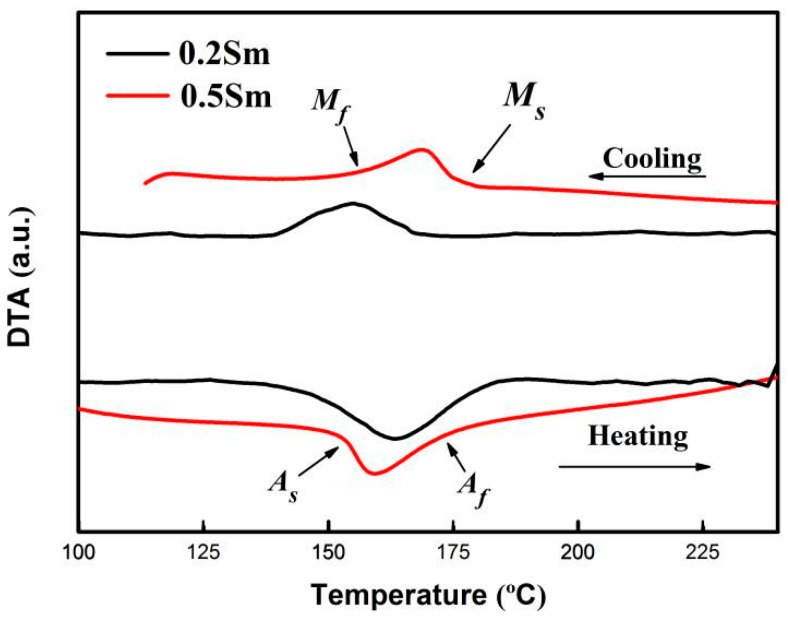
DTA of Cu-13.0Al-4.0Ni-*x*Sm (*x* = 0.2, 0.5) alloys.

**Figure 5 materials-14-04007-f005:**
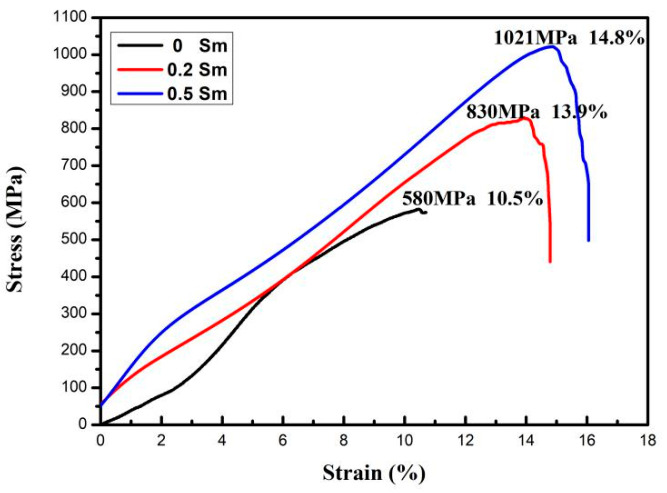
Compressive stress–strain curve of Cu-13.0Al-4.0Ni-*x*Sm (*x* = 0.2, 0.5) alloys.

**Figure 6 materials-14-04007-f006:**
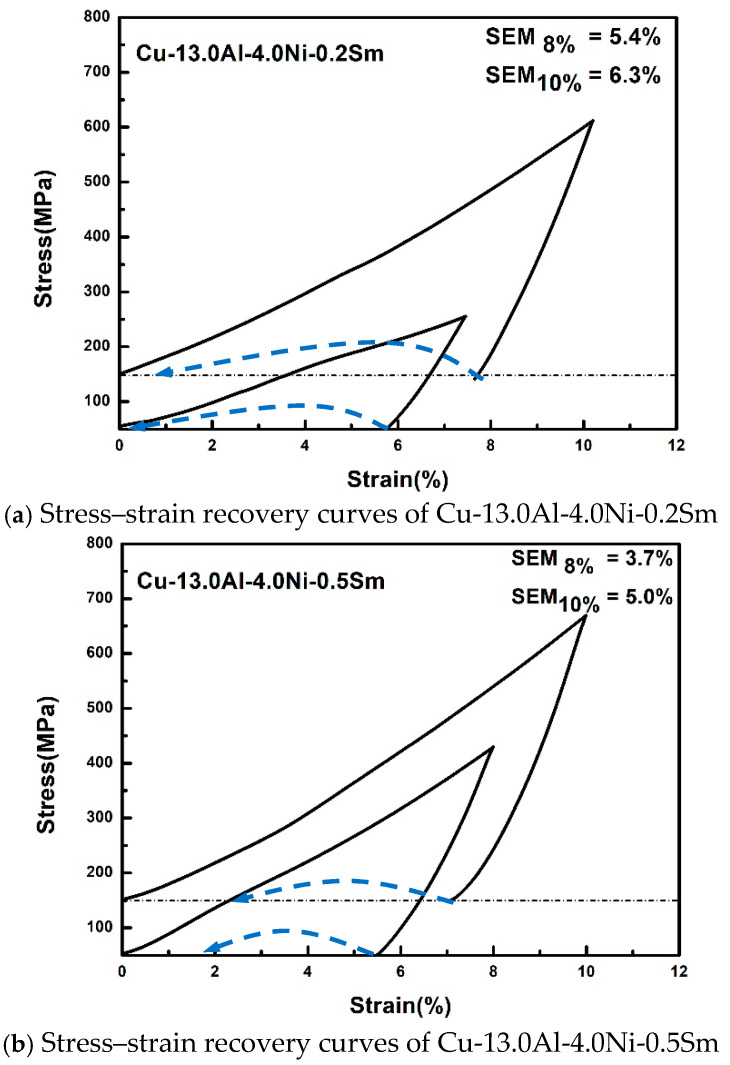
Stress–strain recovery curves of Cu-13.0Al-4.0Ni-*x*Sm (*x* = 0.2, 0.5) alloys under pre-strains of 8% and 10%. The arrow line represents the recovery strain (SME) after heating to 350 °C for 1 min.

**Table 1 materials-14-04007-t001:** Martensitic transformation temperature (°C) of the Cu-13.0Al-4.0Ni-*x*Sm (*x* = 0.2, 0.5) alloy.

Compositions	A_s_	A_f_	M_s_	M_f_
Cu-13Al-4Ni [33]	325	377	229	210
Cu-13Al-4Ni-0.2Sm	142	181	168	139
Cu-13Al-4Ni-0.5Sm	151	176	174	150

**Table 2 materials-14-04007-t002:** Comparison with the mechanical property data and improvement degree of other Cu-Al-Ni alloys doped with a fourth element.

Alloy	Ultimate Compression Strength (Fracture Stress)/MPa (σ_f_)	Strength Improvement (%)	Maximum Strain (Fracture Strain)/% (ε_f_)	Strain Improvement (%)
Cu-13Al-4.0Ni-2.0B [26]	1180	103.4	15	42.9
Cu-11.6Al-3.9Ni-2.5Mn [28]	952	8.1	15	87.5
Cu-14Al-4.5Ni-0.3Ge [30]	1045	65.9	15.2	26.7
Cu-14Al-4.5Ni-0.3Ce [30]	1245	97.6	14.2	18.3
Cu-14Al-4.5Ni-0.3Te [30]	659	4.6	16.2	35.0
Cu-13.0Al-4.0Ni-1.0V [32]	1170	101.7	16.7	59.0
Cu-13.0Al-4.0Ni-0.9Gd [33]	950	63.8	16.5	57.1
Cu-13.0Al-4.0Ni-0.5Nd [34]	940	62.1	18.3	74.3
Present alloy (Cu-13.0Al-4.0Ni-0.5Sm)	1021	76.0	14.8	41.0

## Data Availability

Data sharing not applicable. No new data were created or analyzed in this study. Data sharing is not applicable to this article.

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
