# Peer review of "Effect of Sm Doping on the Microstructure, Mechanical Properties and Shape Memory Effect of Cu-13.0Al-4.0Ni Alloy"

_materials, 2021, doi:10.3390/ma14144007_

Round 1
Reviewer 1 Report
In the article “Effect of Sm doping on the microstructure and properties of Cu-13.0Al-4.0Ni shape memory alloy”, a comparative studies of the microstructure and the mechanical properties of Cu-13.0Al-4.0Ni-xSm alloys with different concentration of Sm element were demonstrated. The Authors tried to reveal the influence of Sm doping on the microstructure and properties of Cu-13.0Al-4.0Ni. However, a few comments should be providing.
- There are no reflexes marked as phase 2H Martensite on the XRD patterns shown in the Figure 2.
- In Figure 3 g, the element Sm is marked as S.
- Figures 4, 5, and 6 do not show the graphs for the Cu-13.0Al-4.0Ni-xSm alloy when x=0. In this case, it is difficult to perform a comparative analysis of the effect of Sm doping on the properties of the alloys.
- The presented manuscript does not contain an analysis and explanation of the dependences of the changes in the mechanical properties of the Cu-13.0Al-4.0Ni-xSm alloy shown in Figures 5 and 6. A more detailed and in-depth discussion involving the work of other researchers is necessary.
Reviewer 2 Report
The authors researched on the effect of Sm doping on the microstructure, mechanical and physical properties of Cu-13.0Al-4.0Ni. The authors particularly study the effect of doping on the grain size and the stress and strain values of the compounds. Overall, the novelty of the study is not specified and there are many technical errors as well that need serious attension. Hence, I would like to reconsider the manuscript after the major revision of the following comments.
1- The title is confusing and incomplete. Especially the phrase "...Cu-13.0Al-4.0Ni shape memory alloy" is making confusion.
2- In the introduction section, briefly explain the novelty of this study. The introduction section is incomplete. What other techniques are contemporarily used to reduce the grain size and improve the mechanical characteristics and shape memory effect of the compound? What other elements are used as a dopant to the compound and why the authors specifically choose Sm for this task? Explain with pertinent references.
3- Provide the CAS numbers of all the chemicals used in the experiments. Also, provide the characteristic details of all the measurement techniques used to characterize the materials and devices.
4- Is it possible to provide more magnified SEM images for better understanding?
5- The stress and strain of the Sm doped compounds should be compared and contrasted with the doped and un-doped compounds presented in recently published papers in tabular form. Use at-least 6 references to compare your results with and establish the fact that your stress and strain values are the best to be considered.
6- Pro vide the limitations to this study and the future directions in the conclusion section.
Reviewer 3 Report
Dear Editor: I would like to express my deep thanks for inviting me to review the manuscript ID: materials-1287408
Title: Effect of Sm doping on the microstructure and properties of Cu-13.0Al-4.0Ni shape memory alloy
Authors: Qimeng Zhang, Bo Cui, Bin Sun, Xin Zhang, Zhizhong Dong and Qingsuo Liu
Comments:
Abstract:
Please rewrite the abstract according to your results.
Introduction:
There is no sufficient information. Rewrite the introduction section and clearly indicate why select this composition with relevant recent work.
Clearly explain the objectives and novelty.
Experimental procedure:
“four elemental elements of Cu, Al, Ni and Sm with purity of 99.99%” it is necessary to add chemical analysis data and supplier information
“The SME and mechanical performance test used cylindrical samples” is it correct?
There is no DTA information.
Results and discussion:
- The quality of Figure is very poor. Replace it by high quality image and add EDS data.
- Authors need to search all peaks and mention the compound name in XRD profile.
- Figure 3 does not provide any information and no discussion.
- In DTA curve please mention the temperature not Mf or Ms
Through out the result and discussion section there is no scientific discussion
Conclusion part:
Please concise the conclusion parts.
RECOMMENDATION
After reviewing the enclosed manuscript for “Materials”, the present manuscript contains some kinds of scientific analysis but it is mandatory required to modify according to the preceding remarks. So, the manuscript can be publication after major revision.
Round 2
Reviewer 1 Report
The authors replied to all comments and made the necessary corrections to the manuscript.
Author Response
Dear reviewer, thanks for your valuable suggestions. Hope my reply can make you healthy!
Reviewer 2 Report
The authors have improved the revised version of the manuscript as per the revision comments. I believe most of the comments were addressed as required. However, the authors have simply stated the fact that their results are better than the cired literature in Table 2 and the explanation is void of any scientific reasoning. I want the authors to provide the scientific reasoning to the claims for readers understanding. The rest of the paper is fine.
Reviewer 3 Report
Authors addressed all my comments in the revised manuscript.
Author Response

(The authors gave the same response as above.)
